# A latitudinal gradient in Darwin's naturalization conundrum at the global scale for flowering plants

Shu-ya Fan [1,16], Qiang Yang [2,3,4,16], Shao-peng Li [1] ✉, Trevor S. Fristoe [2,5], Marc W. Cadotte [6], Franz Essl[7], Holger Kreft [8,9,10], Jan Pergl [11], Petr Pyšek[11,12], Patrick Weigelt [8,10], John Kartesz[13], Misako Nishino[13], Jan J. Wieringa [14] & Mark van Kleunen [2,15]

Darwin's naturalization conundrum describes two seemingly contradictory hypotheses regarding whether alien species closely or distantly related to native species should be more likely to naturalize in regional floras. Both expectations have accumulated empirical support, and whether such apparent inconsistency can be reconciled at the global scale is unclear. Here, using 219,520 native and 9,531 naturalized alien plant species across 487 globally distributed regions, we found a latitudinal gradient in Darwin's naturalization conundrum. Naturalized alien plant species are more closely related to native species at higher latitudes than they are at lower latitudes, indicating a greater influence of preadaptation in harsher climates. Human landscape modification resulted in even steeper latitudinal clines by selecting aliens distantly related to natives in warmer and drier regions. Our results demonstrate that joint consideration of climatic and anthropogenic conditions is critical to reconciling Darwin's naturalization conundrum.

Biological invasions have become ubiquitous across ecosystems worldwide and are a defining characteristic of the Anthropocene[1,2]. So far, over 13,000 plant species have successfully established self-sustaining populations beyond their native ranges (i.e., have become naturalized), and the increase in their numbers worldwide does not show any sign of deceleration[3,4]. These naturalized aliens have resulted in the homogenization of the world's flora[5,6] and exert major environmental, ecological and socioeconomic impacts[7–9]. The distribution of naturalized alien plants is highly uneven globally[4,10], but the determinants of naturalization success and of global geographic patterns therein are not yet fully understood.

[1]Zhejiang Tiantong Forest Ecosystem National Observation and Research Station, Institute of Eco-Chongming, School of Ecological and Environmental Sciences, East China Normal University, Shanghai 200241, China. [2]Ecology, Department of Biology, University of Konstanz, Konstanz 78464, Germany. [3]Institute of Biology, Martin Luther University Halle-Wittenberg, Halle (Saale) 06108, Germany. [4]German Centre for Integrative Biodiversity Research (iDiv) Halle-Jena-Leipzig, Leipzig 04103, Germany. [5]Department of Biology, University of Puerto Rico - Río Piedras, San Juan 00925, Puerto Rico. [6]Department of Biological Sciences, University of Toronto Scarborough, Toronto, ON M1C 1A4, Canada. [7]Division of Bioinvasions, Global Change & Macroecology, Department of Botany and Biodiversity Research, University of Vienna, Vienna 1030, Austria. [8]Biodiversity, Macroecology & Biogeography, University of Göttingen, Göttingen 37077, Germany. [9]Centre of Biodiversity and Sustainable Land Use, University of Göttingen, Göttingen 37077, Germany. [10]Campus-Institut Data Science, Göttingen 37077, Germany. [11]Czech Academy of Sciences, Institute of Botany, Department of Invasion Ecology, Průhonice CZ-25243, Czech Republic. [12]Department of Ecology, Faculty of Science, Charles University, Prague CZ-12844, Czech Republic. [13]Biota of North America Program (BONAP), Chapel Hill 27516 NC, USA. [14]Naturalis Biodiversity Centre, Darwinweg 2, 2333 CR Leiden, Leiden, The Netherlands. [15]Zhejiang Provincial Key Laboratory of Plant Evolutionary Ecology and Conservation, Taizhou University, Taizhou 318000, China. [16]These authors contributed equally: Shu-ya Fan, Qiang Yang. ✉e-mail: spli@des.ecnu.edu.cn

The phylogenetic relatedness between alien species and native species in a recipient region has long been suggested as a predictor of their naturalization success. In 1859, Charles Darwin invoked phylogenetic relationships in two seemingly contradictory hypotheses to explain the naturalization of alien species[11]. In his first hypothesis, now known as the preadaptation hypothesis, Darwin posited that alien species that are closely related to the native species in a regional flora are more likely to naturalize. This is because they are likely to share similar adaptations to the environments in the respective region with their native relatives[11,12]. In his second hypothesis, now known as Darwin's naturalization hypothesis, Darwin posited that the native species in a regional flora could reduce the chances of naturalization for closely related aliens. This is because close relatives should compete more intensely with each other and also because natural enemies of native species might also attack the closely related alien species[11,13]. Together, these two seemingly opposing hypotheses, emphasizing the dominant roles of environmental filtering and biotic interactions, constitute Darwin's naturalization conundrum[14,15] (Fig. 1).

Solving the conundrum has gained increasing attention in recent years, with accumulating evidence in support of both hypotheses[15–20]. The apparent contradiction between the two hypotheses has been partly resolved by considering the spatial scale of studies[14,21]. Evidence for the preadaptation hypothesis is believed to be more detectable at regional scales, where the effects of environmental filtering are more apparent[17]. Conversely, Darwin's naturalization hypothesis is thought to be more applicable at the scale of local communities where species directly interact. However, studies at regional scales have also revealed instances where naturalized species are distantly related to natives[15,16].

These observations could reflect that closely related species are generally more likely to share pathogens, parasites, and herbivores than distantly related species[22–25], and these shared enemies can exert their influence over larger spatial scales. Multiple abiotic and biotic processes, including environmental filtering and shared enemies, could act simultaneously in regulating species naturalization, and their relative strengths could vary considerably even at the regional scale. Therefore, reconciling empirical support for Darwin's two hypotheses across different regional studies remains a major challenge.

We propose that the contradictory findings on Darwin's naturalization conundrum could be clarified by considering the biogeographical context of the study regions (Fig. 1). Darwin posited that in the temperate zone, adaptations are mostly driven by harsh climate, while towards the equator, the importance of biotic interactions should increase (now known as the latitudinal biotic interaction hypothesis[26,27]). We, therefore, expect that support for the preadaptation hypothesis should be stronger in high-latitude regions characterized by cold and highly seasonal environments. In contrast, in low-latitude regions, which usually have relatively stable environments, benign climates, and more intense biotic interactions (e.g., resource competition, herbivory, and disease)[28,29], we expect a higher likelihood of supporting Darwin's naturalization hypothesis. In other words, we predict that with increasing latitude, there should be a decrease in phylogenetic distance between native and naturalized alien plant species (Fig. 1). Indirect support for this hypothesis is provided by studies on native plant assemblages, which have observed an increase in phylogenetic clustering of native plants with latitude[30]. Additionally, some studies found that the phylogenetic structures of

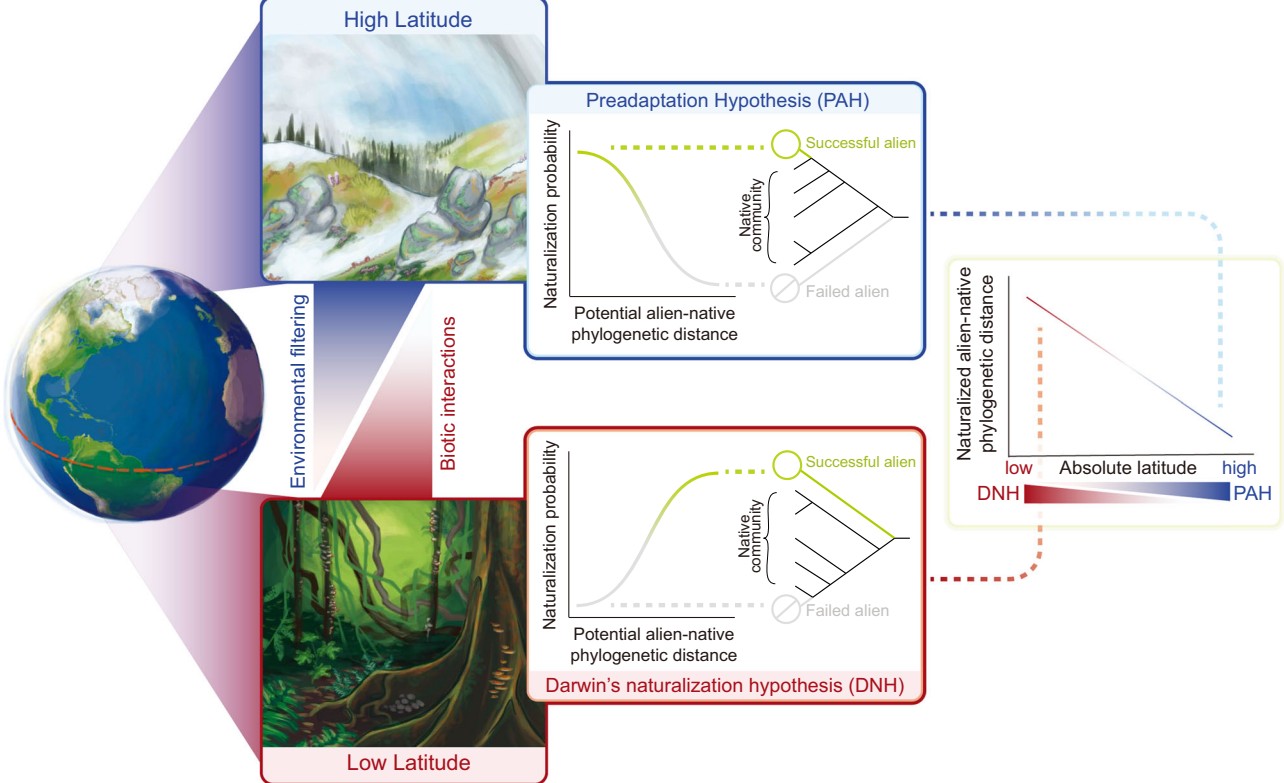

**Fig. 1 | A schematic diagram illustrating the proposed latitudinal gradient in Darwin's naturalization conundrum.** In high-latitude regions characterized by harsh and cold environments, the preadaptation hypothesis (PAH) is expected to receive stronger support due to the predominant role of environmental filtering. Specifically, alien species closely related to native species should have a higher probability of successful naturalization through preadaptation. Conversely, in low-latitude regions with typically warm, stable, and benign environments, Darwin's naturalization hypothesis (DNH) is more likely to be corroborated due to reduced environmental filtering and/or the heightened intensity of biotic interactions. Consequently, alien species distantly related to native species should be more prone to avoid negative interactions and subsequently achieve successful naturalization. Therefore, we predict a decline in the phylogenetic distance between native and naturalized alien species with increasing latitude.

native or alien communities varies with climate[31–33]. These studies, however, did not specifically examine the phylogenetic distance between alien and native species and thus did not directly address Darwin's naturalization conundrum. A notable exception is Park et al.[17], which found that the phylogenetic distance between alien and native plant species increased with temperature and precipitation. However, being restricted to the continental USA, their study did not cover tropical or high-latitude environments and did not explicitly incorporate latitude as a factor. Therefore, a global-scale analysis of regional floras is required to determine whether a latitudinal gradient indeed exists in Darwin's naturalization conundrum and whether it holds true across continents. Such an analysis could potentially clarify some of the seemingly inconsistent findings of previous studies focusing on single regions[15–18].

In this study, to test for latitudinal variation in naturalized-to-native phylogenetic distances, we collated checklists of native and naturalized alien plant species for 487 regions around the world, covering more than 83% of the Earth's ice-free land area. Our dataset included native distributions for 219,520 angiosperms (i.e., flowering plants), naturalized alien distributions for 9531 angiosperms, and phylogenetic information for the full set of species, which represents about 70% of the known global angiosperm flora. Utilizing this data, we calculated the mean pairwise phylogenetic distances (MPD) between naturalized alien and native plant species in each region (Supplementary Fig. 1). We then assessed how MPD varied with latitude and climate variables. Furthermore, given the strong association between species naturalization and human-induced environmental disturbances[34,35], we additionally tested whether the degree of human modification to the landscape influenced the relationship of

naturalized-to-native phylogenetic distance with latitude and climate. Our study reveals a clear global latitudinal pattern of Darwin's naturalization conundrum, and provides evidence that this latitudinal pattern is largely driven by temperature and its seasonality. Moreover, our results demonstrate that these patterns become stronger with increasing human modification of the environment.

## Results and discussion

We found a strong latitudinal gradient in the mean pairwise phylogenetic distances (MPD) between the naturalized alien and native plant species across the globe (Fig. 2a, c). In line with our prediction, naturalized alien species were more distantly related to natives within equatorial regions, and MPD decreased significantly toward the poles (Fig. 2c; $P < 0.001$). To quantify how much higher or lower these MPD values were than expected by chance, we calculated $\Delta$MPD values as the deviations of the observed MPD values from expected values derived from a null model (Supplementary Fig. 1). To ensure the robustness of our findings across different null models, we used six different potential source pools from which the naturalized alien plant species in a recipient region may have derived: (1) a global nonnative species pool (Global nonnative flora), (2) the pool of species with economic uses combined with the global naturalized species pool (Econ. use flora), (3) the global naturalized alien species pool (Global nat.), (4) the pool of species that have naturalized in the continent of the recipient region (Continent nat.), (5) the global pool of species with climatic suitability in the recipient region (Climate nat.), and (6) the pool of climatically suitable species that have also naturalized in the continent of the recipient region (Climate continent nat.; see Methods for details; Supplementary Fig. 2). Irrespective of the choice of the

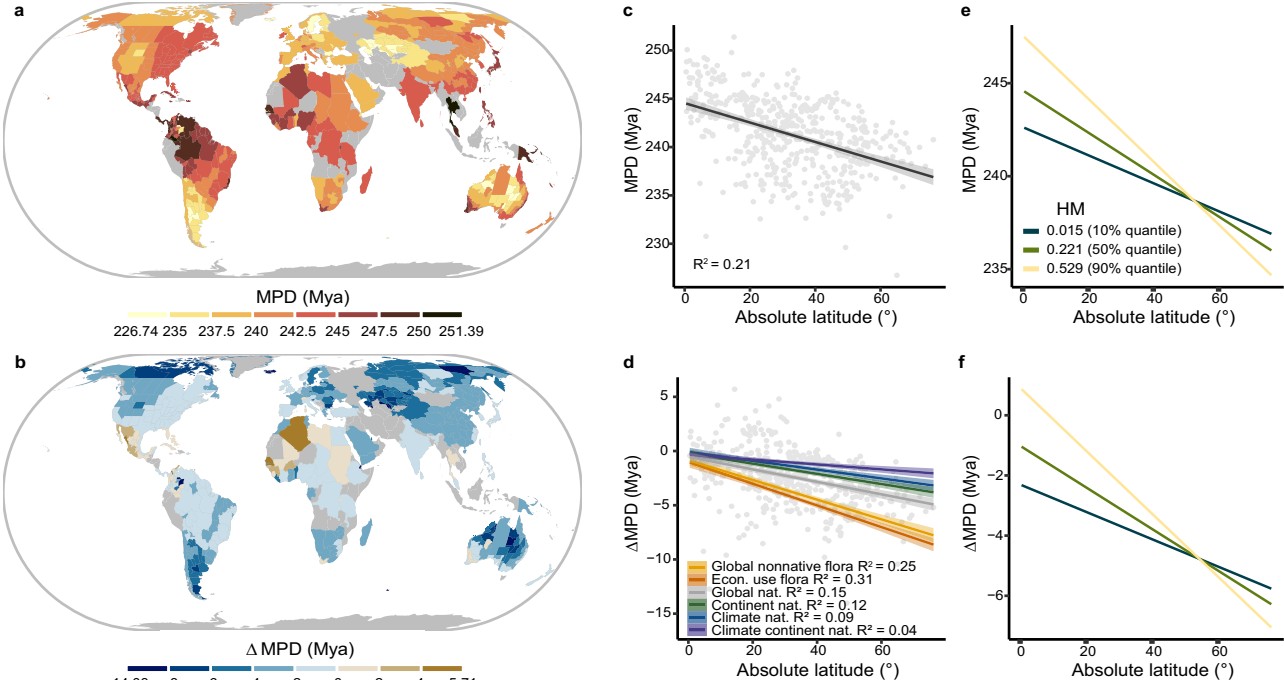

**Fig. 2 | Latitudinal gradient of phylogenetic distance between naturalized alien and native plant species in 487 regions around the world.** Phylogenetic distance was calculated as the mean pairwise phylogenetic distance (MPD, **a**, **c**, **e**) and difference between observed MPD and expected MPD ($\Delta$MPD, **b**, **d**, **f**). Positive values of $\Delta$MPD support Darwin's naturalization hypothesis, whereas negative values support the preadaptation hypothesis. **a**, **b** show the global distribution of MPD and $\Delta$MPD. The $\Delta$MPD values in **b**, **f**, as well as the gray circles in **d**, were generated using the null model based on the global naturalized alien species pool. The colors in **a** and **b** correspond to different MPD and $\Delta$MPD. We used bins with equal intervals, with the exception of the bins for the lowest and highest values, as

there were relatively few observations for those extreme values. Gray areas represent regions without data. In **c** and **d**, the solid lines represent the linear regressions, while the gray shadings indicate the 95% confidence intervals. In **d**, different colors represent different species pools used to calculate $\Delta$MPD (see Methods and Supplementary Fig. 2). **e**, **f** show the interactive effect of the human modification index (HM) and latitude on MPD (two-sided Student's $t$-test: $t$-value = −3.29, $P = 0.001$) and $\Delta$MPD (two-sided Student's $t$-test: $t$-value = −3.15, $P = 0.002$). The lines represent the model predictions when fixing HM at its 10%, 50% and 90% quantiles. All relationships are statistically significant ($P < 0.001$), and all $R^2$ values have been adjusted.

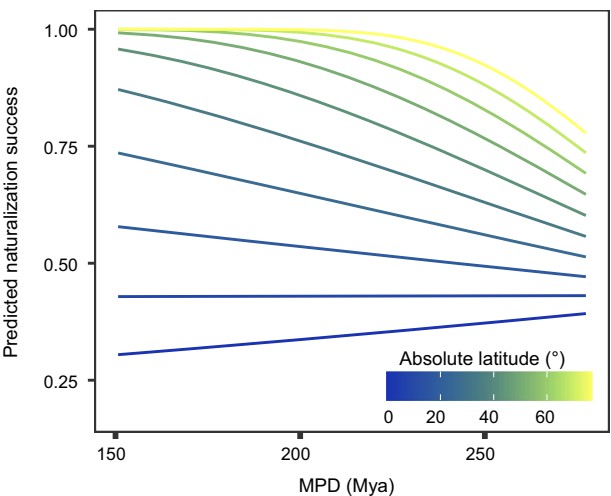

**Fig. 3 | The effect of phylogenetic distance on species naturalization probability across latitude.** Phylogenetic distance was calculated as the MPD between each species from the global naturalized alien flora (9,531 species) and all native species in each region ($n = 4,403,923$ observations). The partial relationships of naturalization probability with MPD are shown when fixing latitude for different values. The partial relationships are based on a binomial generalized mixed linear model with a fixed effect MPD (two-sided Wald $Z$-test: $z$-value $= -5.27$, $P < 0.001$), absolute latitude (two-sided Wald $Z$-test: $z$-value $= 3.65$, $P < 0.001$) and their interaction (two-sided Wald $Z$-test: $z$-value $= -3.39$, $P = 0.001$). The colors of the lines indicate the latitudes.

species pools, ΔMPD consistently decreased with increasing latitude (Fig. 2d; $P < 0.001$ for all models). The phylogenetic distances between the naturalized alien and native plant species were significantly higher than expected in some low-latitude regions (e.g., Algeria and some Mexican states; Fig. 2b and Supplementary Figs. 3 and 4), consistent with the prediction of Darwin's naturalization hypothesis. Conversely, in most mid- and high-latitude regions (e.g., Northern Europe and Canada), MPD was significantly lower than expected (Fig. 2b and Supplementary Figs. 3 and 4), which supports the preadaptation hypothesis. As previous studies have found, native flora usually have a phylogenetic clustering tendency under pronounced environmental filtering at high latitudes[30]. Our findings of a latitudinal gradient in phylogenetic distances between the naturalized alien and native plant species suggest that similar mechanisms contribute to the successful establishment of closely related alien plants in these environments.

In addition to analyzing the average MPD and ΔMPD values between all naturalized alien and native plant species in a region, similar patterns were obtained when we assessed the relationship between naturalization success (yes, no) versus phylogenetic distances of each of the 9531 naturalized alien plant species in our dataset to the natives in each region where they are not native (Fig. 3). At high latitudes, the naturalization probability of an alien species was higher in regions with closely related native floras, whereas, at low latitudes, the reverse was true (Fig. 3). Therefore, both the analysis at the regional flora and individual species level showed that the probability of observing evidence for the preadaptation hypothesis increases from the equator to the poles.

We further tested whether the geographic patterns were explained by global climatic gradients. We first performed a principal component analysis (PCA) on 19 biologically relevant climate variables[36]. The first PCA axis ($PC_{Temp}$), explaining 45.7% of the climatic variation, primarily represented temperature and its seasonality, with high values indicating warm, non-seasonal environments (Supplementary Table 1). Notably, $PC_{Temp}$ exhibited a significant negative correlation with latitude (Pearson's $r = -0.943$, $P < 0.001$, Supplementary Fig. 5a). The second PCA axis ($PC_{Prec}$), explaining 31.7% of the

climatic variation, mainly captured information related to precipitation and its seasonality, with high values representing humid regions with non-seasonal precipitation (Supplementary Table 1). Consistent with our expectations, MPD and the six ΔMPDs significantly increased with $PC_{Temp}$ and $PC_{Prec}$ (Fig. 4). In cold and seasonal regions (such as Northern Europe and Canada) or arid regions (such as Central Asia, Interior Australia, and the Southern Cone of South America), we observed a higher likelihood of naturalization for alien species closely related to natives (Fig. 2b and Supplementary Figs. 3 and 4). This finding aligns with the results of a previous study[17], indicating that environmental filtering plays a more important role in determining species' naturalization in harsher climates. In warmer and non-seasonal regions (such as western Mexico), naturalized alien species were generally more distantly related to natives (Fig. 2b and Supplementary Figs. 3 and 4), which may reflect the relatively weak abiotic constraints imposed by benign climates. Instead, intense interactions with enemies and competitors may be more prevalent in these typically biodiverse regions, which tend to hinder the naturalization of closely related species[27,37–40].

The latitudinal gradient of naturalized-to-native phylogenetic distances was evident in most continents, including Africa, Asia, Europe, Northern America, and Southern America (Supplementary Fig. 6). However, Australasia showed a reversed latitudinal gradient (Supplementary Fig. 6), which is consistent with its known latitudinal gradient of plant diversity[41]. This reversal is likely due to the pronounced increase in precipitation with latitude across Australasia compared to the other continents (Supplementary Figs. 5b and 7b). Aridity appears to act as the primary environmental filter for species naturalization in this continents. Consistently, we found that both MPD and the ΔMPDs decreased in regions with low $PC_{Prec}$ across Australasia (Supplementary Fig. 7). This is in line with the idea that Australasia, characterized by the second lowest precipitation levels after Antarctica, exhibits biodiversity patterns strongly shaped by variation in precipitation[42]. Therefore, the reversed gradient in Australasia highlights the necessity of conducting studies spanning multiple continents, as they not only confirm the general patterns of species naturalization at a macroecological scale but also reveal exceptions that can provide insights into underlying mechanisms.

While macroclimatic conditions can strongly constrain naturalization success, it is important to note that these climatic variables, along with other potentially influential environmental factors[43], are also subject to modification by human activities in various ways and to different degrees across the globe. Therefore, to investigate the potential influence of human modification on the observed latitudinal and climatic patterns, we further explored the interactions between the human modification index (HM) and latitude, as well as climate variables, on MPD and ΔMPD. The HM index quantifies the extent to which humans have changed land cover[44]. Our analysis revealed that higher values of HM strengthened the latitudinal pattern (Fig. 2e, f; $P < 0.001$ for the interaction between HM and latitude in all models) and resulted in steeper gradients of MPD and ΔMPD with $PC_{Temp}$ (Fig. 4b, d). The steeper gradients primarily arose from higher MPD and ΔMPD values at low-latitude, warm regions. This observation could be attributed to the relatively recent intense human modifications in tropical regions that have created many novel environments[34,45]. Such disturbances may particularly benefit alien species from distantly related temperate clades that have preadapted to anthropogenic landscapes through their long histories in human-modified environments (e.g., in Europe and Asia)[46]. While the precise mechanisms warrant further investigation, our results suggest that with increasing human modification of environments worldwide, the latitudinal gradient of naturalized-to-native phylogenetic distances might steepen.

We also found that in regions with low levels of human modification, increasingly arid conditions (i.e., lower $PC_{Prec}$ values) were associated with lower naturalized-to-native phylogenetic distances

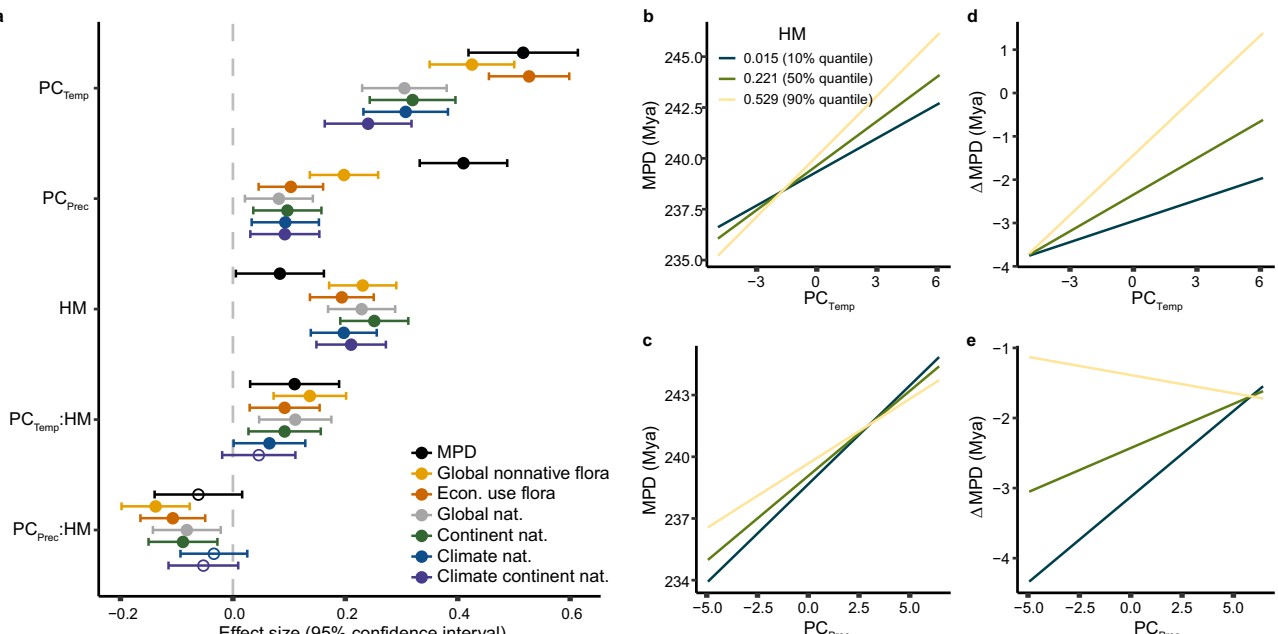

**Fig. 4 | Effects of climate and human environmental modification on phylogenetic distance between naturalized alien and native species. a** shows the effect sizes of the climate variables and the human modification (HM) index and their interactions on MPD and the ΔMPDs, estimated by linear mixed-effects models. Different colors represent different species pools used to calculate ΔMPD (see Methods and Supplementary Fig. 2). The circles represent the coefficient estimates and the error bars represent the 95% confidence interval (n = 487 regions). Solid circles denote statistically significant effects, while open circles represent non-significant effects. In **b**–**e**, the partial relationship of MPD and ΔMPD with PC_Temp and PC_Prec are shown when fixing HM at its 10%, 50%, and 90% quantiles. In **d** and **e**, ΔMPD values were generated from the global naturalized alien species pool.

(Fig. 4c). This result is again consistent with environmental filtering as a primary constraint on the naturalization of alien species in harsh climates. However, with increasing human modification, the positive relationship between MPD and PC_Prec became shallower and was even reversed to negative in regions with the highest human modification (Fig. 4c, e). Possibly, this reflects that human modifications, such as the construction of artificial water bodies and other forms of water management, as well as irrigation of agricultural land, locally alleviate water limitation in arid regions. These human modifications could facilitate the establishment of distantly related alien species that would otherwise be filtered out by the arid natural conditions.

Our findings resonate with previous studies indicating that both environmental filtering and biotic interactions act simultaneously and thus jointly shape naturalization success[15,47]. Our analysis revealed that in some tropical regions, the observed naturalized-to-native phylogenetic distance was larger than expected. These results thus support Darwin's naturalization hypothesis, indicating that in these tropical regions, the influence of biotic interactions may outweigh environmental filtering. The reverse pattern was observed in most other regions and particularly in the cold, seasonal environments found at high latitudes, indicating that environmental filtering predominates in most regions. The prevalence of negative ΔMPD values in our study is likely attributable to our focus on regional floras rather than local communities. Indeed, previous studies have found that negative relationships between alien-to-native phylogenetic distance and naturalization success were more likely found for larger spatial units (e.g., administrative units, like in our study), whereas positive relationships are more likely, but not exclusively, observed at local spatial scales (e.g., within vegetation plots), where alien and native plants are more likely to directly compete[16,17]. At the spatial resolution of our study, the alien and closely related native plant species may not necessarily directly interact with each other. Nevertheless, it is important to recognize that they can still affect each other indirectly through various biotic interactions, such as shared pathogens, herbivores,

pollinators, and seed-dispersers. It should also be noted that the occurrence or absence of species at a regional scale is an emergent outcome of local processes. The regional lists of native species provide an indication of the potential compositions of local communities and the probability of an alien encountering closely or distantly related species. While our data do not allow for direct observation of local interactions, we can still detect a signal of their influence at regional scales. In fact, our results align with the prediction of the latitudinal biotic interaction hypothesis, which suggests that biotic interactions are more pronounced at lower latitudes[27].

While our analyses encompassed extensive data covering most of the terrestrial world, we acknowledge the potential for future advancements. Our study encountered limitations in data coverage, particularly in parts of Southeast Asia and Africa (Fig. 2), emphasizing the importance of cautious interpretation to account for potential biases. It is therefore crucial to prioritize efforts in obtaining more comprehensive species lists, particularly for naturalized aliens, in these underrepresented regions. With the continuous expansion of public datasets, there is an increased opportunity to explore latitudinal and climatic patterns of alien-to-native phylogenetic relatedness, not only across regional floras but also across local communities (i.e., vegetation plots) and across various spatial scales. Furthermore, it is worth noting that our phylogenetic tree, although informative, contains several polytomies at the genus level due to the current unavailability of a fully resolved global flora phylogeny. However, it has been shown that the use of genus-level phylogenetic trees instead of fully resolved ones have minimal impact on measuring patterns of phylogenetic relatedness among collections of species that contain different families or genera[48–52]. This is because the deep branches of the phylogenetic tree have a strong influence on the overall patterns. Nevertheless, we acknowledge that our analyses could be improved with the advent of a more accurate species-level phylogenetic tree.

While Darwin clearly referred to the importance of the phylogenetic relatedness between aliens and native species in a region[11], he

evoked phylogenetic relatedness as an integrative surrogate for similarity in key functional traits and niche preferences between alien and native species. This assumption is reasonable considering the prevalence of niche conservatism[53]. The current lack of comprehensive trait data and methods of linking the different traits to niche differences pose limitations[54]. Future global studies should strive to incorporate functional distances and niche similarities between alien and native species in order to enhance our understanding of Darwin's conundrum. Moreover, as it was recently shown that the effect of alien-to-native phylogenetic distance might vary among the different stages of the invasion process[19,55], future studies should also consider those different stages under varying geographic and environmental contexts. Together with important insights from previous studies, our findings of latitudinal and climatic gradients in the effect of phylogenetic distance on naturalization and the effects of human environmental modification on these relationships will help to solve this 160-year-old Darwin's naturalization conundrum.

## Methods

### Native and naturalized alien angiosperm floras

We extracted regional lists of naturalized alien and native angiosperms (i.e., flowering plant species) from the Global Naturalized Alien Flora (GloNAF) database[56] and the Global Inventory of Floras and Traits (GIFT) database[57,58], respectively. As the most comprehensive global database of naturalized alien plants, GloNAF contains lists of naturalized alien vascular plant species for various regions (e.g., countries, states, provinces, islands) that together cover most of the world's ice-free terrestrial surface[10,56]. GIFT is a compilation of floras and checklists of predominantly native plant species with an indication of their native status for more than 300,000 species across nearly 3000 regions with near-global coverage[57,58]. We selected regions present in both GloNAF and GIFT. Additionally, we merged some GloNAF regions to match the higher-level administrative divisions in GIFT regions and vice versa. To further increase the spatial coverage of our analysis, we also collected checklists of native species for some GloNAF regions from other data sources (see Supplementary Table 2 for details), when the native species lists were not yet included or incomplete in GIFT.

Regions with data on native and naturalized alien plants vary largely in their size. Because the study scale may affect the relationship between phylogenetic relatedness and species naturalization[17], we reduced this potential effect by restricting our analyses to regions larger than 5000 km². Small mainland regions were merged into the country that they belong to, and small oceanic islands were excluded from our analysis. However, to assess the sensitivity of the analyses to the region-size threshold, we also conducted additional tests using thresholds of 0, 1000, and 10,000 km², which confirmed that patterns were consistent regardless of the threshold (Supplementary Table 3). Furthermore, including region area in our statistical model did not change the statistical significance and signs of the effect sizes of the explanatory variables that we were interested in (Supplementary Fig. 8). We therefore only report results without region area as an explanatory variable in the main text. Our selection criteria based on the 5000 km² threshold resulted in 487 regions, which in total cover more than 83% of the world's terrestrial ice-free surface.

Because regional checklists of non-angiosperms are usually not as complete as those of angiosperms, we restricted our analysis to angiosperms. We harmonized all taxonomic names of both naturalized alien and native species using the R package Taxonstand v.2.4[59], according to The Plant List (http://www.theplantlist.org/), which was used as a taxonomic reference list for both the GloNAF[56] and GIFT[57,58] databases. We excluded hybrids and assigned infraspecific taxa to the binomial species level. If infraspecific taxa were standardized to two different accepted binomial names, they were considered to be separate species, and their native or alien status was assigned accordingly.

If multiple infraspecific taxa were standardized to the same binomial species name, the species was considered native if at least one of the infraspecific taxa was classified as native. Species with an uncertain native/alien status or with a conflicting status (including 1286 of 1,156,207 records in total), i.e., being native to a region according to GIFT but being naturalized to the same region according to GloNAF, were assigned as native species. However, the results remained consistent when we assigned them as naturalized aliens (Supplementary Fig. 9). After cleaning the data, our final dataset encompassed distributions of 9531 naturalized alien species and 219,520 native species across 487 regions, which represent ~70% of the global angiosperm flora (Supplementary Fig. 10).

### Phylogenetic analyses

To calculate the phylogenetic distances between species, we constructed a phylogenetic tree for the global flora using accepted names from The Plant List. As a basis for this tree, we used the phylogeny "ALLMB" constructed by Smith and Brown (2018)[60], which is to date one of the most comprehensive trees for seed plants. We first standardized the taxon names of this tree with The Plant List using the package Taxonstand[59], and pruned the tree to only include taxa with accepted names in The Plant List. Some plants with accepted names in The Plant List were not found in the tree, and we therefore added them manually. Specifically, for species that were missing but had congeneric species present in the tree (7.2% of species), we grafted those species to the roots of the corresponding genera. When the entire genus was missing from the tree (1.3% of species), we grafted the species to the roots of the families they belong to. We then pruned the global flora tree to only include the 224,593 species analyzed in this study (including all native angiosperms, naturalized alien angiosperms, and economically useful angiosperms mentioned below). Some species in our mega tree species were added as polytomies of their genera or family, while others are fully resolved at the species level. This approach might introduce potential biases when examining latitudinal and climatic gradients in naturalized-to-native phylogenetic distances. Therefore, we also created a tree with a lower but more even phylogenetic resolution across regions. This was achieved by adding all species to the roots of the corresponding genera or, when the entire genus was missing from the tree, to the roots of the families they belong to. We found that the use of this less-resolved tree yielded similar results (Supplementary Fig. 11). This finding supports recent studies showing that phylogenetic metrics derived from mega-phylogenies resolved only at the genus level and those derived from fully resolved species-level trees are nearly perfectly correlated (Pearson's correlation coefficients ranged from 0.966 to 1.000)[51,52]. The reason for this high correlation is that the assembled communities typically contain distantly related species across different families or genera that traverse deep nodes. Thus, fine-tuning the relative distance among species within the same genus has minimal impact on phylogenetic distance metrics.

For each region, the phylogenetic distance between naturalized aliens and native species can be calculated as the mean pairwise phylogenetic distance (MPD) and as the mean nearest taxon distance (MNTD)[61,62]. MPD represents the average of the pairwise phylogenetic distances between naturalized aliens and the native species in a region (Supplementary Fig. 1), and could therefore capture both shallow and deep branching in the phylogenetic tree[62,63]. In contrast, MNTD estimates the average of the phylogenetic distances of each naturalized alien species with its closest native relative in the focal region and is more strongly influenced by patterns towards the tips of the tree[62,63]. As it is difficult to achieve good resolution at the tips of the large mega-phylogeny with numerous polytomies, we opted to only use MPD in our analysis.

To assess whether the naturalized alien species were more closely or more distantly related to the native ones than expected by random

chance, we adopted a null-model approach. This involved comparing the observed MPD between naturalized alien and native species to the distribution of simulated distances that were drawn randomly from a potential alien species source pool. Specifically, we conducted 1000 random simulations by keeping the native species composition and the number of naturalized alien species in a region unchanged but randomly drawing the identities of the naturalized alien species from the respective source pool. Hence, this approach allowed us to account for the different potential source pools of alien species, the likelihood of introduction to a region, and the probability of successful naturalization based on the climatic suitability of the focal regions. For each region and null model, we initially calculated the standardized effect size of MPD (SES.MPD) as (observed MPD – mean of the expected MPDs) divided by the standard deviation of the expected MPDs. While SES.MPD provides information about whether the observed MPD value of a region deviates significantly from the expected values, it is less suitable for comparisons among regions because of SES.MPD values are sensitive to variation in species richness[64,65]. To avoid this, it has been suggested to use rarefaction[64], but because the variance in MPD values expected from randomizations is higher in species-poor regions, rarefaction unnecessarily reduces the certainty of estimates for many regions[65]. Therefore, we instead calculated ΔMPD as the difference between observed MPD and the mean of the expected MPDs (Supplementary Fig. 1) and used the inverse of the variance of the distribution of expected MPDs as a weight in the statistical models to account for uncertainty (see the section 'Statistical analyses' below). Nevertheless, if we analyzed SES.MPD instead of ΔMPD, the results were very similar (Supplementary Figs. 12 and 13). Positive values of ΔMPD (or SES.MPD) indicate that naturalized alien species are more distantly related to native species in a region than expected by chance, aligning with Darwin's naturalization hypothesis. Conversely, negative values indicate that naturalized alien species are more closely related to native species, supporting the preadaptation hypothesis. All phylogenetic distance indices were calculated using the R package picante v.1.8.2[66].

How a potential source pool of naturalized alien species is defined can affect the output of the null models and the interpretation of the results[67]. To assess whether the patterns we found were robust, we considered six different potential pools of alien species (Supplementary Fig. 2). These pools of potentially introduced species for the focal region differed in size. The first species pool included all species not native to the focal region (i.e., the global nonnative species pool, global nonnative flora). As particularly alien species with economic uses, such as for human food and animal fodder, are likely to be introduced and become naturalized[68], we used as a second species pool of a focal region all non-native species that have a known economic use. However, because that would exclude naturalized species without known economic use, we also included in this species pool all other species from the global naturalized flora (i.e., the pool of economic-use species combined with the global naturalized species pool, econ. use flora). Data on species with economic uses was extracted from the World Checklist of Useful Plant Species[69]. As a third species pool, we used all species that have become naturalized in at least one region worldwide (i.e., the global naturalized alien species pool, global nat.). While it is likely that some naturalized alien species have been introduced into the great majority of the world, this might not apply to all naturalized alien species. Therefore, we used, as a fourth species pool, all species that have naturalized in at least one region of the biogeographical continent that the focal region is part of (i.e. the pool of species that have naturalized in the same continent as a given recipient region, continent nat.). This assumes that when a species is naturalized in at least one region of a continent, it is more likely to have also been introduced to all other regions of that continent. For this null model, we divided the world into eight continents using the Taxonomic Databases Working Group continent scheme (TDWG, including Africa, Asia, Europe, South

America, North America, Antarctica, the Pacific Islands, and Australasia, https://www.tdwg.org/standards/wgsrpd/). We combined the TDWG continental regions Asia-Temperate and Asia-Tropical into Asia because of the short latitudinal range in Asia-Tropical.

The fifth species pool considered the climatic suitability of the naturalized alien species for the focal region (i.e., the global pool of naturalized species with climatic suitability in a recipient region, climate nat.). This approach accounts for the fact that humans are more likely to intentionally introduce climatically suitable plant species than climatically unsuitable ones[70,71]. Additionally, this null model controls for the fact that species from across temperate continents tend to be more closely related than those from across tropical continents[72]. To predict the potential geographic distribution of each species in the global naturalized flora, we employed maximum entropy modeling (MaxEnt v.3.4.1) using the R package dismo v.1.3.5[73]. This modeling was based on bioclimatic variables from both its native and naturalized ranges, and relied on six climate variables extracted from WorldClim at 2.5 arc-min resolution. These variables included mean diurnal range, temperature seasonality, the maximum temperature of the warmest month, precipitation seasonality, precipitation of the wettest quarter, and precipitation of driest quarter, which are known to be related to species distributions. To ensure the reliability of our models, we examined pairwise Pearson's correlations between these climate variables and found them to be below 0.7. This indicates that collinearity among the variables should have a negligible effect on subsequent model estimations and predictions[74]. Ideally, climatic niche modeling would rely on known point-occurrence data instead of regional occurrence data. Although some widely naturalized alien species (i.e., invasive species) have extensive point-occurrence records, comprehensive point-occurrence data for all naturalized alien species at a global scale is currently unavailable. Consequently, we used coarse occurrence data (i.e., regional occurrence data) to construct species distribution models. To do this, the global map was rasterized into 50 km × 50 km grid cells, with the assumption that a species that occurs in a region resides in each grid cell of that region. Additionally, we upscaled the six bioclimatic variables to match the resolution of the regional occurrence data. To ensure the reliability of our models in predicting global distribution of all naturalized alien species, we firstly focused on species with more than 15 occurrences, as that number is considered to be sufficient for running a species distribution model[75]. For each naturalized alien species, we then allocated 80% of the distribution data for model training and 20% of the distribution data for testing the model's performance. We retained models with an area under the curve (AUC) greater than 0.7 (considered a valid model)[76]. Using these valid models, we proceeded to predict the climatic suitability of each species in every grid cell, based on climatic data in the 487 regions. Subsequently, we extracted the potentially climatically suitable regions for each naturalized alien species by using the maximum training sensitivity plus specificity threshold. Specifically, if a region has at least one grid cell with suitability not lower than the threshold, then we postulate the species to be climatically suitable for this region. Finally, we defined the climatically suitable species pool for each focal region by assuming that only the species whose potential distribution overlaps with the focal region could naturalize there. The sixth and last species pool was even more strict and combined the criteria used to make the fourth and fifth species pools. In other words, the sixth species pool assumed that for a focal region, only the species that are both climatically suitable and have also naturalized in the continent of the focal region have a chance to naturalize there (i.e., the pool of climatically suitable species that have also naturalized in the continent of the recipient region, climate continent nat.).

## Climate variables and human modification index
We considered climate variables and human modification of the landscape as potential drivers of global patterns in the naturalized-to-

native phylogenetic distance. For climate variables, we used all 19 bioclimatic variables from WorldClim[36] at 2.5 arc-min resolution. To account for collinearity among the 19 bioclimatic variables, we performed a PCA on the centered and scaled bioclimatic variables. Due to the skewed distribution of some variables, we used various transformations to approximate normal distributions (see Supplementary Table 1) by using the R package normalizer v.0.1.0[77]. The first two PC axes ($PC_{Temp}$ and $PC_{Prec}$) accounted for 77.8% of the variation, where $PC_{Temp}$ mainly reflects the temperature and its seasonality, and $PC_{Prec}$ mainly reflects precipitation and its seasonality (Supplementary Table 1). Then, we calculated $PC_{Temp}$ and $PC_{Prec}$ for each region as the mean value across all grid cells in that region. These two variables are also strongly correlated with the global aridity index (AI), and potential evapotranspiration (ET)[78] (Supplementary Fig. 14). Therefore, we only considered $PC_{Temp}$ and $PC_{Prec}$ in our analyses, which provide independent indices of temperature and precipitation effects.

To characterize the influence of anthropogenic stressors, we incorporated the global human modification index (HM). HM is strongly positively correlated with human population density[79] (Supplementary Fig. 14), but is an index that much more comprehensively quantifies human-caused changes of terrestrial lands (including changes due to human settlement, agriculture, transport, energy production and electrical infrastructure)[44]. The HM is available at 1-km² resolution and ranges from 0 (low human impact) to 1 (high human impact)[44]. The average HM for each region was calculated as the mean value across all grid cells in that region.

## Statistical analyses

To describe the general pattern of naturalized-to-native phylogenetic distance along the latitudinal gradient, we first used linear regressions to test whether there were statistically significant relationships ($P < 0.05$) of latitude with MPD and the six ΔMPDs, each based on a different null model. To explore how human modification of the environment affects the latitudinal gradient of phylogenetic distance, we additionally ran multiple linear regression models of MPD and ΔMPD on latitude, HM, and their interaction. We used the R package visreg v.2.7.0[80] to visualize the effect of the significant interaction between HM and latitude ($P < 0.05$; Fig. 2e, f). For plotting, we set the HM values to the 10%, 50%, and 90% quantiles of the HM values across all regions.

We further performed multiple linear mixed-effects models to assess the relative importance of $PC_{Temp}$, $PC_{Prec}$ and their interactions with HM on MPD, and the six ΔMPDs. To account for differences among the TDWG continents, we included the continent as a random intercept. To account for uncertainty in the estimation of ΔMPD, we used the inverse of the variance of the distribution of expected MPDs as a weight in the ΔMPD models. Linear mixed-effects models were performed in the R package lme4 v.1.1.28[81]. The standardized effect sizes of the predictor variables in the above models were calculated using the R package effectsize v.0.6.0.1[82]. Again, we used the R package visreg v.2.7.0[80] to visualize the effect of the significant interaction ($P < 0.05$) between HM and climate variables ($PC_{Temp}$ and $PC_{Prec}$) on MPD and ΔMPD based on the global naturalized species pool. Specifically, we plotted the partial relationship of climate variables with MPD and ΔMPD for the 10%, 50%, and 90% quantiles of HM. $R^2$ in all models has been adjusted, and all statistical analyses were conducted in R v.4.0.3[83].

The analyses described in the preceding section calculate for each focal region the average of the MPD values across the naturalized alien species in that region, which means that region is the unit of analysis. As an alternative analysis, we calculated MPD of each of the 9531 naturalized alien species in our dataset to the natives in each of the 487 regions where the species is not native. Then we ran a binomial generalized linear mixed model, using the R package lme4 v.1.1.28[81], across all species and regions to test how the naturalization

success of a species in a region relates to its MPD to the natives in that region, the latitude of the region and their interaction. To account for non-independence of observations for the same species and the same region, we included species identity and region identity as random factors. Furthermore, as species might vary in their responses to MPD and latitude, we also included random slopes of species with regard to MPD, latitude, and their interaction. Since the binomial response variable had many more zeros than ones, we used the complementary log-log link function[84]. For visualizing the statistically significant interactions of MPD and latitude ($P < 0.05$), we fixed latitude at several fixed values to depict the predicted relationship between species naturalization success and MPD at different latitudes.

Considering the variation in species inventory completeness among regions, with notably better documentation of species in highly developed areas, we conducted additional tests to identify potential biases that this variation might introduce to the latitudinal patterns we found. Specifically, we first focused on the regions (467 regions) where the estimated completeness of naturalized species lists exceeded 50%, based on completeness-estimate scores from the GloNAF database[56]. In this subset of regions, we re-analyzed the linear relationship between MPD and latitude. Second, we obtained the average native species inventory completeness percentages in 430 regions from Dawson et al. (2017)[1]. Then we developed the linear regression model of MPD with latitude for these 430 regions, incorporating the average inventory completeness percentage as a weighting variable. This weighting approach assigns greater weight to regions with higher completeness levels. As these additional analyses still revealed a latitudinal cline in naturalized-to-native phylogenetic relatedness (Supplementary Table 4), we only show the results based on all regions in the main text.

## Reporting summary

Further information on research design is available in the Nature Portfolio Reporting Summary linked to this article.

## Data availability

The data generated in this study have been deposited in the figshare (https://doi.org/10.6084/m9.figshare.20055611.v5)[85]. The GloNAF database together with the shapefile that was used to produce the maps have been published in a data paper[56]. The GIFT database is accessible via the GIFT R-package (https://CRAN.R-project.org/package=GIFT)[58].

## Code availability

The code for the analyses and figures are available at figshare (https://doi.org/10.6084/m9.figshare.20055611.v5)[85].

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

## Acknowledgements

We thank Andrey Kupriyanov and Aleksandr L. Ebel for providing plant distribution data. This research was supported by the National Natural Science Foundation of China (NSFC3222051, NSFC31971553) to S.P.L. and the Fundamental Research Funds for the Central Universities. M.v.K. and Q.Y. thank the German Research Foundation DFG (grant 264740629), F.E. appreciates funding by the Austrian Science Foundation FWF. P.P. and J.P. were supported by EXPRO grant no. 19-28807X (Czech Science Foundation) and long-term research development project RVO 67985939 (Czech Academy of Sciences).

## Author contributions

S.P.L. and M.v.K. came up with the original idea, S.Y.F. and Q.Y. prepared and analyzed the data and wrote the first manuscript draft with major inputs from S.P.L., T.S.F., M.W.C. and M.v.K. F.E., H.K., J.P., P.P., P.W., J.K., M.N. and J.J.W. provided data and contributed to the writing.

## Competing interests

The authors declare no competing interests.
