## [Peer review file · Nature Communications]

REVIEWER COMMENTS

Reviewer #1 (Remarks to the Author):

Review for Nature Communications

Title: A latitudinal gradient in Darwin's naturalization conundrum at the global scale

General comments

With a large-scale, macroecological approach, this study tests whether an important challenge in invasion biology, Darwin's naturalization conundrum, can be solved when taking into account large-scale global differences in latitude, temperature and precipitation, and human land use. The striking result is that indeed, most of the expectations are met: At high latitudes and harsh environments, alien and native plant species are closer related to each other, indicating advantages of pre-adaptedness, whereas at low latitude and less harsh conditions, aliens and natives are less closely related, indicating advantages of being dissimilar and thus more novel for competitors and enemies. Human landscape modification even enhances the pattern.

This is a remarkable study. The authors report on a very meaningful pattern that strongly aligns with conceptual considerations. It seems they found evidence for general rules about naturalization success that are linked to likely mechanistic explanations – a very rare case in ecology. It is remarkable that even the exception they found (Australasia) turned out to match the provided explanations very well when looking closer.

As far as I can judge, the data sources are of high quality. It seems the authors took great care in assembling a data set that is highly suited to test the research question. The approach they took to assembling it is sophisticated and appropriate. The overall approach is well-chosen. I appreciate that the authors test for effects of temperature and precipitation in addition to latitude, and the use of different plant species pools for null models. I especially liked the idea of having a species pool that includes economically used species, since this is data I have not very often seen being used for these kinds of analyses.

The methods and results are presented and visualized in a clear and understandable way. The presentation of the results in maps is very appealing and approachable. Also, figures S1 and S2 included in the supplement are quite helpful for clarifying the approach of the analyses. However, the underlying conceptual framework, Darwin's naturalization conundrum, is pretty complex, and I wonder if there would be some space left for a conceptual diagram, helping readers not familiar with these ideas to understand the underlying considerations.

All in all, the study results seem robust, and highly reliable, and the conclusions are considerate and do not contain any over-interpretation. The text is well-written and there is a good line of argument guiding the reader.

To conclude, this study is a rare example where empirical results neatly match major hypotheses, and an impressive demonstration of the power of large-scale analysis of global datasets combined with conceptual depth. Usually, I am quite a critical reviewer (and not generally in favor of macroecological studies), but in this case, I did not find any major points of critique. I strongly recommend publication.

Note: I have reviewed a previous version of this manuscript for a different journal, and this report builds on my previous review. In my point of view, this current version is much stronger than the previous version I read, with improved analyses and language.

Minor comments

Line 82: "be" is missing ("can be clarified")

Line 119: I think it is confusing to call the global alien species pool 'global flora' because to me this sounds as if native species were included.

Line 197 to 200: Something is wrong with this sentence, please revise.

Lines 440-453: The final data set giving alien and native species per region, compiled and cleaned as specified in the method section, is a highly valuable resource for future studies. Do you plan on publishing this as well?

Reviewer: Tina Heger

ORCID <https://orcid.org/0000-0002-5522-5632>

Reviewer #2 (Remarks to the Author):

The authors test Darwin's naturalization conundrum at a global scale, and find a latitudinal gradient in how closely related naturalized angiosperm species are to native ones. They also examine how this trend may be mediated by human landscape modification. The manuscript is clearly written, but lacks some detail – this may be partially due to the fairly short format it was written in. The study largely confirms previous studies that have found that the relatedness between species is greater in harsher climates (which underlies the overall latitudinal gradient; e.g., Qian et al. 2013; LI et al. 2014; Escoriza & Ruhí 2014; Park et al. 2020; Kusumoto et al. 2021; Zu et al. 2023) and albeit at a larger scale. Likewise, the effects of human landscape modification on phylogenetic community patterns have been investigated, but at smaller scales and more heterogeneous results (e.g., Cui et al 2019; Yang et al 2022). The larger scale of this study is both its strength and weakness. It lends support to previous findings at larger scales and provides an interesting, if not surprising, global-scale view of how human dislocated species may integrate into regional flora. However, this comes at a cost – the underlying data are not as detailed/accurate, the coarse spatial resolution cannot account for biotic interactions among taxa, and thus the results do not advance our knowledge as to why we observe these patterns. There is a proliferation of large-scale correlative analyses based on data that others have collected aimed at testing hypotheses on global scales. Though such studies have value, they often do not add new knowledge to the field, especially when the gathered data are not made available in their entirety. Along these lines, the study is somewhat descriptive and does not introduce novel insights, new methods, or new data. Again, the strength and novelty of this study are in its large-scale and the assemble global dataset of native and naturalized plants, but without making the underlying data fully accessible, it is less useful. Please make data openly available and not "by request".

The examination of human landscape modification effects is a potentially interesting addition that has not been investigated as often. However, I think there needs to be some tests to rule out the possibility that the observed patterns are due to lack of resolution and/or biases in the data. For instance, highly developed nations with extensively modified landscapes are often the places where both native and non-native species are best documented. These nations also simultaneously engage in large amounts of international trade and travel, in effect, increasing propagule pressure of exotic species and their chances of establishment (Dawson et al 2017). These developed nations mostly reside in higher-latitude temperate regions. Similarly, it is possible that higher-latitude regions with similar floras (e.g., North America and northeast Asia) engage in more biotic exchange than they do with lower-latitude regions, as suggested by some of the authors' previous work (van Kleunen et al. 2015).

Some of the methodological choices need more justification/explanation. For instance, why is >5000 km² the threshold for analysis? Why was The Plant List used for taxonomic harmonization when it has been static since 2013 and replaced? Several of the 19 bioclimatic variables used to model species distributions are highly correlated and should not be included in the same models. Sampling biases

associated with the data sources and their potential effects on the results should be discussed in more detail.

Also, while the authors should be commended for their testing of multiple null models, I am curious as to why native species were not included as candidates when shuffling taxa. At the end of the day Darwin's naturalization conundrum is a hypothesis of community assembly and thus should consider the insertion of any species that has the means to physically reach a community, regardless of its native status. For example, the fact that native species X is not present in community/region A despite occurring nearby for millennia, while non-native species Y from the other side of the world has inserted itself over the last 50 years tells us something.

Figure legends and sizing could be better – for instance, it was very difficult to see the inset map in Figure 1. Also, figure captions could be clearer. Individual descriptions of each panel may help. How were the colors binned? The bins do not seem to be equal in size, both in the main figures and the supplementary figures.

Table S2: Where do the IDs for each region come from?

Line 82: There's a word missing. Think it should be "can be clarified"

Line 92: Another word missing. Should be "would allow us to test"

Line 106-107: The statement is slightly confusing/misleading. The study examines whether the degree of human modification affects the relationship between latitude (and climate) and the relatedness between native and naturalized species. It does not, however, examine if/how human modification alters the biotic/abiotic environment. Also, I do not think the study actually tests "whether the degree of human modification of the landscape has changed the latitudinal gradient" – it only shows a significant interaction term.

Line 166: "could also" seems strange here. Latitude is often seen as a proxy for climatic factors and does not generally have a direct effect on biodiversity.

Line 259: How many taxa had conflicting native statuses? How were native vs non-native subspecies treated? Do the results change if taxa with conflicting status were considered non-native?

Line 285: Genus-level phylogenies may be sufficient in certain circumstances but such a blanket statement is improper and can be dangerous. Further, the test that was conducted – making all genera polytomies and redoing analyses – does not demonstrate that the lack of phylogenetic resolution does not affect the results. What it is showing is that the results from the tree used in the study, which is not fully resolved nor fully supported by molecular data, are not largely different from a tree with worse resolution. It does not prove that the results from the used tree are congruent with those that would be generated from a more robust, resolved phylogeny supported by molecular data. Perhaps the authors could limit their analyses to regions for which they have full phylogenetic coverage and see if the results still stand?

Line 332: The rationale for limiting a species pool to species with known economic uses is not clear, especially when the focal taxon itself does not have a known use, and needs to be better fleshed out/justified. Why not just limit this pool to species that have been introduced somewhere?

Line 365: The modeling of species distributions with such coarse data are troubling. 250 km² can comprise extreme amounts of environmental heterogeneity – e.g., coastal invaders can become inland invaders of the desert or alpine invaders. Perhaps some comparisons between these crude models and

those fit to actual point occurrence data could be compared for taxa with enough data to support this approach – many (if not most) invasive species have large amounts of occurrence data available. Also, all model parameters should be provided to ensure reproducibility.

Qian, H., Zhang, Y., Zhang, J. and Wang, X. (2013), Phylogenetic structure of angiosperm tree assemblages. *Global Ecology and Biogeography*, 22: 1183-1191. <https://doi.org/10.1111/geb.12069>

LI, X. H., ZHU, X. X., Niu, Y., & Sun, H. (2014). Phylogenetic clustering and overdispersion for alpine plants along elevational gradient in the Hengduan Mountains Region, southwest China. *Journal of Systematics and Evolution*, 52(3), 280-288.

Escoriza, D., & Ruhí, A. (2014). Macroecological patterns of amphibian assemblages in the Western Palearctic: Implications for conservation. *Biological Conservation*, 176, 252-261.

Park, D. S., Feng, X., Maitner, B. S., Ernst, K. C., & Enquist, B. J. (2020). Darwin's naturalization conundrum can be explained by spatial scale. *Proceedings of the National Academy of Sciences*, 117(20), 10904-10910.

Kusumoto, B., Kubota, Y., Shiono, T., & Villalobos, F. (2021). Biogeographical origin effects on exotic plants colonization in the insular flora of Japan. *Biological Invasions*, 23(9), 2973-2984.

Zu, K., Zhang, C., Chen, F., Zhang, Z., Ahmad, S., & Nabi, G. (2023). Latitudinal gradients of angiosperm plant diversity and phylogenetic structure in China's nature reserves. *Global Ecology and Conservation*, 42, e02403.

Yang, Y., Wei, C., Xiao, L., Zhong, Z., Li, Q., Wang, H., & Wang, W. (2022). Effects of urbanization on woody plant phylogenetic diversity and its associations with landscape features in the high latitude northern hemisphere region, Northeast China. *Science of The Total Environment*, 838, 156192.

Cui, Y. C., Song, K., Guo, X. Y., van Bodegom, P. M., Pan, Y. J., Tian, Z. H., ... & Da, L. J. (2019). Phylogenetic and functional structures of plant communities along a spatiotemporal urbanization gradient: Effects of colonization and extinction. *Journal of Vegetation Science*, 30(2), 341-351.

Dawson, W., Moser, D., Van Kleunen, M., Kreft, H., Pergl, J., Pyšek, P., ... & Essl, F. (2017). Global hotspots and correlates of alien species richness across taxonomic groups. *Nature Ecology & Evolution*, 1(7), 0186.

Van Kleunen, M., Dawson, W., Essl, F., Pergl, J., Winter, M., Weber, E., ... & Pyšek, P. (2015). Global exchange and accumulation of non-native plants. *Nature*, 525(7567), 100-103.

Responses to the comments

Responses to the Reviewer 1's comments

General comments

With a large-scale, macroecological approach, this study tests whether an important challenge in invasion biology, Darwin's naturalization conundrum, can be solved when taking into account large-scale global differences in latitude, temperature and precipitation, and human land use. The striking result is that indeed, most of the expectations are met: At high latitudes and harsh environments, alien and native plant species are closer related to each other, indicating advantages of pre-adaptedness, whereas at low latitude and less harsh conditions, aliens and natives are less closely related, indicating advantages of being dissimilar and thus more novel for competitors and enemies. Human landscape modification even enhances the pattern.

This is a remarkable study. The authors report on a very meaningful pattern that strongly aligns with conceptual considerations. It seems they found evidence for general rules about naturalization success that are linked to likely mechanistic explanations – a very rare case in ecology. It is remarkable that even the exception they found (Australasia) turned out to match the provided explanations very well when looking closer.

RESPONSE: We sincerely thank the reviewer for this very positive evaluation.

As far as I can judge, the data sources are of high quality. It seems the authors took great care in assembling a data set that is highly suited to test the research question. The approach they took to assembling it is sophisticated and appropriate. The overall approach is well-chosen. I appreciate that the authors test for effects of temperature and precipitation in addition to latitude, and the use of different plant species pools for null models. I especially liked the idea of having a species pool that includes economically used species, since this is data I have not very often seen being used for these kinds of analyses.

RESPONSE: We thank the reviewer for the appreciation of the assembled data and the approach we used. We are pleased to see that this reviewer notes the value of including the economic species pool, as we also discuss this species pool below in response to a comment by Reviewer 2.

The methods and results are presented and visualized in a clear and understandable way. The presentation of the results in maps is very appealing and approachable. Also, figures S1 and S2 included in the supplement are quite helpful for clarifying the approach of the analyses. However, the underlying conceptual framework, Darwin's naturalization conundrum, is pretty complex, and I wonder if there would be some space left for a conceptual diagram, helping readers not familiar with these ideas to understand the underlying considerations.

RESPONSE: This is a very good suggestion, and we now included a diagram illustrating the

underlying conceptual framework (see Fig. 1)

All in all, the study results seem robust, and highly reliable, and the conclusions are considerate and do not contain any over-interpretation. The text is well-written and there is a good line of argument guiding the reader.

To conclude, this study is a rare example where empirical results neatly match major hypotheses, and an impressive demonstration of the power of large-scale analysis of global datasets combined with conceptual depth. Usually, I am quite a critical reviewer (and not generally in favor of macroecological studies), but in this case, I did not find any major points of critique. I strongly recommend publication.

Note: I have reviewed a previous version of this manuscript for a different journal, and this report builds on my previous review. In my point of view, this current version is much stronger than the previous version I read, with improved analyses and language.

RESPONSE: We thank the reviewer for this very positive evaluation.

Minor comments

Line 82: “be” is missing (“can be clarified”)

RESPONSE: Corrected (Line 82)

Line 119: I think it is confusing to call the global alien species pool ‘global flora’ because to me this sounds as if native species were included.

RESPONSE: We agree, and now refer to this species pool as the ‘global nonnative species pool’.

Line 197 to 200: Something is wrong with this sentence, please revise.

RESPONSE: We apologize for this weird sentence. We now revised it to “Our results showed that in some of the tropical regions, the observed naturalized-to-native phylogenetic distance was larger than expected. These results thus support Darwin’s naturalization hypothesis, indicating that in these tropical regions, the influence of biotic interactions may outweigh environmental filtering” (Lines 222-225).

Lines 440-453: The final data set giving alien and native species per region, compiled and cleaned as specified in the method section, is a highly valuable resource for future studies. Do you plan on publishing this as well?

RESPONSE: Based on this comment and a similar comment of Reviewer 2, we decided to make the entire dataset available (Lines 508-509). The dataset, encompassing the lists of alien and native species for all 493 regions considered in our analysis, is now openly accessible through Figshare (Data.species.distribution.493.Rdata, <https://doi.org/10.6084/m9.figshare.20055611.v4>).

Responses to the Reviewer 2's comments

The authors test Darwin's naturalization conundrum at a global scale, and find a latitudinal gradient in how closely related naturalized angiosperm species are to native ones. They also examine how this trend may be mediated by human landscape modification. The manuscript is clearly written, but lacks some detail – this may be partially due to the fairly short format it was written in.

RESPONSE: We sincerely appreciate the reviewer's acknowledgment that the manuscript is clearly written. In response to the reviewer's valuable feedback provided below, we have carefully addressed these specific comments by incorporating additional details into the revised version of the manuscript.

The study largely confirms previous studies that have found that the relatedness between species is greater in harsher climates (which underlies the overall latitudinal gradient; e.g., Qian et al. 2013; Li et al. 2014; Escoriza & Ruhí 2014; Park et al. 2020; Kusumoto et al. 2021; Zu et al. 2023) and albeit at a larger scale. Likewise, the effects of human landscape modification on phylogenetic community patterns have been investigated, but at smaller scales and more heterogeneous results (e.g., Cui et al 2019; Yang et al 2022).

RESPONSE: Thank you for bringing up these relevant studies on taxonomic distinctness (Escoriza et al., 2014) or phylogenetic clustering/dispersion of native species (Zu et al., 2023), alien species (Kusumoto et al., 2021), or native and alien species jointly (Qian et al., 2013; Li et al., 2014), and we now cite several of them (Lines 92-99). However, we would like to point out that these studies mainly focused on assessing phylogenetic community structures within communities themselves, none of them specifically examined the relatedness of alien species to native floras, which is the core aspect of Darwin's naturalization conundrum. One notable exception is Park et al. (2020), which directly analyzed alien-native phylogenetic distances within the US and how they relate to spatial scale. A novel finding of Park et al. (2020), as we mention in the main text (Lines 98-101, 170-172), is that alien-native phylogenetic distances were smaller at larger spatial extents and in harsher climates. However, the study did not explicitly investigate the latitudinal gradient of Darwin's naturalization conundrum, possibly due to the limited latitudinal range covered by the US alone. Therefore, to the best of our knowledge, our study is the first to propose the existence of a latitudinal gradient in Darwin's naturalization conundrum and provides empirical evidence for this phenomenon at both continental and global scales.

Another novel aspect of our study is the consideration of multiple null models (species pools) to assess whether the alien-native phylogenetic relatedness is smaller or larger than expected. Because it is impossible to know all species that actually have been introduced into the focal region and to recognize that the definition and size of the potential source pool may affect the outcomes, we used multiple species pools based on different assumptions of which species have most likely been introduced. By using these different species pools, we aim to enhance our understanding of the role of phylogenetic relatedness in naturalization success, while accounting for potential phylogenetic biases introduced by specific introduction preferences.

The larger scale of this study is both its strength and weakness. It lends support to previous findings at larger scales and provides an interesting, if not surprising, global-scale view of how human dislocated species may integrate into regional flora. However, this comes at a cost – the underlying data are not as detailed/accurate, the coarse spatial resolution cannot account for biotic interactions among taxa, and thus the results do not advance our knowledge as to why we observe these patterns.

RESPONSE: We agree with the reviewer that the coarse spatial resolution has its limitations. However, we respectfully disagree with the assertion that the results therefore do not add new knowledge or important insights. Our study reveals a previously unreported pattern—the latitudinal gradient in the relatedness between alien and native species at the global scale. This finding, we believe, significantly contributes to our understanding of the distribution and integration of naturalized species in regional floras.

In regard to discerning mechanisms, we point out that this is not only a challenge in large/global scale analyses such as ours, but in any observational/correlational study, also the ones that use a smaller spatial resolution (Escoriza et al., 2014; Kusumoto et al., 2021; Park et al., 2020; Qian et al., 2013). However, as commended by Reviewer 1, we have clear mechanistically based hypotheses on the expected latitudinal and environmental patterns in alien-native phylogenetic distance. While the nature of our data precludes us from definitively concluding that the discussed mechanisms drive the patterns, the fact that empirical data clearly support expectations at this scale is worth pointing out, and we believe that our results have the potential to prompt future studies to explore these mechanisms using diverse and complementary data and approaches.

We sincerely acknowledge that at the spatial resolution of our study, the alien and closely related native plant species do not necessarily interact directly. However, we recognize that even without direct interactions, species can influence each other through various biotic interactions, such as shared pathogens, herbivores, pollinators and seed-dispersers. Furthermore, the occurrence or absence of species at a regional scale is influenced by processes playing out at local scales. While we cannot observe local biotic interactions directly in our data, we can detect signals of their influence at regional scales. In fact, our results align with the prediction of the latitudinal biotic interaction hypothesis, which suggests that biotic interactions are more pronounced at lower latitudes (Schemske et al. 2009). We therefore consider the potential increase in biotic interactions at low latitudes as a plausible explanation for the novel latitudinal gradient pattern we identified.

Thank you for the valuable feedback and we have discussed these points more clearly in the revised manuscript (Lines 233-244).

There is a proliferation of large-scale correlative analyses based on data that others have collected aimed at testing hypotheses on global scales. Though such studies have value, they often do not add new knowledge to the field, especially when the gathered data are not made available in their entirety.

RESPONSE: We would like to emphasize that our study adds new knowledge to the field by

proposing and empirically demonstrating the presence of a latitudinal gradient in Darwin's naturalization conundrum at a global scale. This novel finding offers a potential explanation for the mixed results obtained in previous regional-scale studies conducted at different latitudes.

In consideration of the comments of both reviewers, we have conferred with contributing authors (who have all spent significant time and effort in collecting, gathering, aligning and organizing the various underlying datasets) and decided to make the data openly available. The dataset includes the lists of both alien and native species for all 493 regions analyzed in our study, and it can be accessed through Figshare. (Data.species.distribution.493.Rdata, <https://doi.org/10.6084/m9.figshare.20055611.v4>).

Along these lines, the study is somewhat descriptive and does not introduce novel insights, new methods, or new data. Again, the strength and novelty of this study are in its large-scale and the assemble global dataset of native and naturalized plants, but without making the underlying data fully accessible, it is less useful. Please make data openly available and not "by request".

RESPONSE: We respectfully disagree with the notion that our study lacks novel insights, new methods or new data. While it is true that previous studies have reported on phylogenetic community structures along latitude, our study goes beyond this by proposing and empirically testing how the relatedness between alien and native species changes along latitudinal gradients. To the best of our knowledge, this is the first study to investigate the latitudinal gradient of Darwin's naturalization conundrum.

Furthermore, we have introduced new methods in our study, particularly through the utilization of multiple and innovative species pools (e.g., a species pool that includes economically used species). We believe that these novel approaches will greatly benefit future studies on Darwin's naturalization conundrum, facilitating comparisons and selection of appropriate species pools for analysis.

Moreover, in response to the suggestions raised by the reviewers, we have made the decision to make our complete dataset openly accessible. By openly sharing our global dataset on native and naturalized plants, we aim to facilitate further research and encourage validation and exploration of related topics.

The examination of human landscape modification effects is a potentially interesting addition that has not been investigated as often. However, I think there needs to be some tests to rule out the possibility that the observed patterns are due to lack of resolution and/or biases in the data. For instance, highly developed nations with extensively modified landscapes are often the places where both native and non-native species are best documented. These nations also simultaneously engage in large amounts of international trade and travel, in effect, increasing propagule pressure of exotic species and their chances of establishment (Dawson et al 2017). These developed nations mostly reside in higher-latitude temperate regions. Similarly, it is possible that higher-latitude regions with similar floras (e.g., North America and northeast Asia) engage in

more biotic exchange than they do with lower-latitude regions, as suggested by some of the authors' previous work (van Kleunen et al. 2015).

RESPONSE: We agree that species are more likely to be best documented in highly developed regions. To address the potential bias introduced by the completeness of species inventories, we have conducted two additional analyses. First, we reanalyzed the data using only regions estimated to have over 50% completeness of their naturalized species lists, based on completeness-estimate scores from the GloNAF database (van Kleunen et al. 2019). The majority of regions in our study (470 out of 493) met this criterion, ensuring a reasonable level of completeness. We reanalyzed the data using only these regions and found that the latitudinal cline in alien-native phylogenetic relatedness persisted. Second, for the 437 regions that overlapped with the study by Dawson et al. (2017), we incorporated the average inventory completeness percentage as a weighting variable in our analyses. By giving more weight to regions with higher completeness, we aimed to address potential biases. This analysis also confirmed the presence of a latitudinal cline in alien-native phylogenetic relatedness (Response Letter Table 1).

Response Letter Table 1 | Results on the linear regression between MPD and latitude (1) without accounting for data completeness, (2) for the subset of regions for which the naturalized species lists were judged to have over 50% completeness (the completeness score in the GloNAF database; van Kleunen et al. 2019), and (3) for the subset of regions that overlapped with Dawson et al. (2017) and for which the inventory completeness percentage was used as weighting variable.

Models	Regions' number	Estimate	Std. Error	t value	Pr(> t)	Adjusted R-squared
(1)	493	-0.117	0.009	-12.589	<0.0001	0.243
(2)	470	-0.103	0.009	-11.22	<0.0001	0.210
(3)	437	-0.109	0.011	-10.193	<0.0001	0.191

In van Kleunen et al. (2015), we have indeed shown that certain parts of the world, such as North America and Temperate Asia, have exchanged more species with one another than with others. This might partly reflect that some continents are climatically more similar, and we considered this by using a global as well as a continental pool of naturalized species that are climatically suitability to the focal region (mentioned in Lines 412-413). Moreover, in another paper (van Kleunen et al. (2020), we showed that many of the biotic exchange patterns among different parts of the world can be explained by the species that we cultivate for their economic uses. We addressed this by also having a pool of species with economic uses.

Some of the methodological choices need more justification/explanation. For instance, why is >5000 km² the threshold for analysis? Why was The Plant List used for taxonomic harmonization when it has been static since 2013 and replaced? Several of the 19 bioclimatic variables used to model species distributions are highly correlated and should not be included in the same models. Sampling biases associated with the data sources and their potential effects on the results should be discussed in more detail.

RESPONSE: According to this thoughtful comment, we carefully revised the method section of our manuscript to address the concerns raised.

- (1) Regarding the threshold of region size, we now clarify more elaborately that we excluded regions $<5000 \text{ km}^2$ to minimize the variation in region size, as smaller regions may be subject to greater variation and potential sampling biases (Lines 291-294). Additionally, we conducted sensitivity analyses using different region size thresholds (0, 1000, 5000, and 10,000 km^2) to assess the robustness of our findings. We found that all thresholds consistently showed a negative relationship between phylogenetic distance and latitude (Lines 295-298; Supplementary Table 3). In addition, we also tested whether the relationship between MPD and latitude varied by region sizes. For this, we divided the data into three subsets based on the 33% and 67% quantiles of the region-size distribution. Analyses of each of the three resulting subsets shows that they all reveal negative MPD vs latitude relationships (Response Letter Figure 1). These findings indicate the robustness of the alien-native MPD-latitude relationship, regardless of the chosen region-size thresholds.

Response Letter Figure 1 | The relationship of MPD with latitude for separate analyses of the one third of regions with the smallest areas ($\log_{10}(\text{Area}) < 4.407$), the one third of regions with intermediate areas ($4.407 \leq \log_{10}(\text{Area}) < 5.139$) and the one third of regions with the largest areas ($\log_{10}(\text{Area}) \geq 5.139$). Pearson correlation coefficient and p value (two sided) are shown.

- (2) Regarding the taxonomic harmonization, we clarified that The Plant List was used because both the GIFT database (Weigelt et al., 2020) and the GloNAF database (van Kleunen et al., 2019) were harmonized using The Plant List as a reference (Lines 306-307).
- (3) Regarding the bioclimatic variables, the reviewer is right that some of the 19 variables are highly correlated. Although we believe this should not necessarily be problematic for predictive models, we now ran the MaxEnt models using only six of the bioclimatic variables (including mean diurnal range, temperature seasonality, maximum temperature of warmest month, precipitation seasonality, precipitation of wettest quarter, and precipitation of driest quarter), all with correlations <0.7 (details are provided in lines 416-423). We then recalculated the

expected MPDs based on the climatically suitable global naturalized alien species pool (climate nat.) and the climatically suitable continental naturalized alien species pool (climate continent nat.). Although our results show that the results are consistent when including all 19 variables or only the six variables with low collinearity (Response Letter Figure 2), we adopted the reviewer's suggestion and now included the results based on the six climatic variables in the main text. We updated the results in the manuscript accordingly (Figs. 2d, 4a, and Supplementary Figs. 4, 6, 7, 8, 11, 12, 13).

Response Letter Figure 2 | Comparison of Δ MPD values calculated using all 19 bioclimatic variables or a subset of six bioclimatic values with $r < 0.7$ for (a) the climatically suitable global naturalized alien species pool and (b) the climatically suitable continental naturalized alien species pool.

(4) Regarding potential biases, we have expanded the discussion to provide a more thorough exploration of the limitations of the data (Lines 245-252). While we acknowledge these limitations, we believe that the data sources we utilized are of high quality on a global scale, and that these potential limitations are unlikely to significantly impact our main conclusions. According to the reviewer's suggestion, we highlighted that incorporating well-resolved species occurrence data at regional and local scales, as well as species-level molecular phylogenies and trait data, would be valuable for future research on Darwin's naturalization conundrum (Lines 249-252, 258-261, 265-269). Unfortunately, these approaches are not currently feasible with the available data. We have included these considerations and limitations in the revised manuscript to provide a more comprehensive overview (Lines 233-269)

Also, while the authors should be commended for their testing of multiple null models, I am curious as to why native species were not included as candidates when shuffling taxa. At the end of the day Darwin's naturalization conundrum is a hypothesis of community assembly and thus should consider the insertion of any species that has the means to physically reach a community, regardless of its native status. For example, the fact that native species X is not present in community/region A despite occurring nearby for millennia, while non-native species Y from the

other side of the world has inserted itself over the last 50 years tells us something.

RESPONSE: We acknowledge that we may not fully understand the reviewer's comment, but we can see two possible interpretations:

(1) If the reviewer suggests shuffling both native and alien species in each region, making all species being randomly selected from the larger species pool, we recognize that such an approach would be a conventional test of phylogenetic community structures and community assembly. This line of research, as acknowledged by some of the studies cited by this reviewer, addresses important questions. However, it is crucial to note that the hypotheses being investigated in this context are fundamentally different from our study on Darwin's naturalization conundrum. As highlighted by Thuiller et al. (2010) in their influential review paper, the null model employed for testing Darwin's naturalization conundrum should specifically break down the phylogenetic relationship between alien and native species, while preserving the phylogenetic relationships between native species within the recipient regions. In our study, the question tested by the null model is why a particular alien species successfully naturalized in a recipient region, rather than another alien species from the available pool. Therefore, the implementation of the randomization procedure in our study must not change the evolutionary and ecological mechanisms that led to the current structure of native communities.

(2) If the reviewer suggests including species native to the continent but not native to the focal region itself in the potential species pool, we already recognize the point made. Such species are currently already included in the global non-native species pool of a region. However, to address this more explicitly, we have now created a species pool for each region that encompasses all species native or naturalized to the continent but not native to the specific region. For this null model, like for all our other null models, we still find a negative Δ MPD vs latitude relationship (Response Letter Figure 3). Considering that we are unsure whether the reviewer intended this specific type of null model and since such species are already included in the global non-native species pool of a region, we did not include this new null model in the manuscript. However, we are willing to incorporate it if the editor believes it is necessary.

Thanks for this comment and please feel free to correct us if we misunderstood any aspects of the point.

Response Letter Figure 3 | Latitudinal gradient of Δ MPD based on the null model using a species pool consisting of all species (native and alien) occurring in the focal region's continent, but that are not native to the focal region.

Figure legends and sizing could be better – for instance, it was very difficult to see the inset map in Figure 1. Also, figure captions could be clearer. Individual descriptions of each panel may help. How were the colors binned? The bins do not seem to be equal in size, both in the main figures and the supplementary figures.

RESPONSE: We thank the reviewer for the useful suggestions regarding the figures. Following the recommendation, we have relocated the inset map from Fig. 1 to Supplementary Fig. 3, and we now provide detailed descriptions for each panel in the figure captions (Lines 734-740, 758). In the revised manuscript, we also clarified that we used bins of equal interval sizes, except for the lowest and highest values due to the relatively few observations in these larger bin intervals (Lines 737-739).

Table S2: Where do the IDs for each region come from?

RESPONSE: We have clarified in the revised manuscript that the IDs for each region correspond to the IDs used in the open-access GloNAF database (see <https://idata.idiv.de/DDM/Data/ShowData/257>).

Line 82: There's a word missing. Think it should be "can be clarified"

RESPONSE: corrected (Line 82).

Line 92: Another word missing. Should be "would allow us to test"

RESPONSE: Thank you for catching this. This sentence has been rephrased to "Therefore, a global-scale analysis of regional floras is required to determine whether a latitudinal gradient indeed exists in Darwin's naturalization conundrum and whether it holds true across continents" (Lines 101-103).

Line 106-107: The statement is slightly confusing/misleading. The study examines whether the

degree of human modification affects the relationship between latitude (and climate) and the relatedness between native and naturalized species. It does not, however, examine if/how human modification alters the biotic/abiotic environment. Also, I do not think the study actually tests “whether the degree of human modification of the landscape has changed the latitudinal gradient” – it only shows a significant interaction term.

RESPONSE: We agree, and this sentence has been rephrased to “Furthermore, given the strong association between species naturalization and human-induced environmental disturbances, we additionally tested whether the degree of human modification to the landscape influenced the relationship of naturalized-to-native phylogenetic distance with latitude and climate.” (Lines 114-117).

Line 166: “could also” seems strange here. Latitude is often seen as a proxy for climatic factors and does not generally have a direct effect on biodiversity.

RESPONSE: We agree. This sentence has been removed from the main text.

Line 259: How many taxa had conflicting native statuses? How were native vs non-native subspecies treated? Do the results change if taxa with conflicting status were considered non-native?

RESPONSE: In the revised manuscript, we clarified that there were 1,278 taxa-by-region combinations out of 1,177,267 that had conflicting native statuses (Lines 312-313). We also added a figure to the supplements showing that alien-native phylogenetic distance estimates are not affected by whether the species with conflicting status are considered to be natives or aliens (Supplementary Fig. 9). Regarding native vs non-native subspecies (or other infraspecific taxa), if they were standardized to different accepted binomial names, they were considered as separate species, and we considered the native vs non-native status of those species. If the infraspecific taxa were standardized to the same binomial species name, they were treated as native species in our study. We now mention this explicitly in the Methods (lines 308-311).

Line 285: Genus-level phylogenies may be sufficient in certain circumstances but such a blanket statement is improper and can be dangerous. Further, the test that was conducted – making all genera polytomies and redoing analyses – does not demonstrate that the lack of phylogenetic resolution does not affect the results. What it is showing is that the results from the tree used in the study, which is not fully resolved nor fully supported by molecular data, are not largely different from a tree with worse resolution. It does not prove that the results from the used tree are congruent with those that would be generated from a more robust, resolved phylogeny supported by molecular data. Perhaps the authors could limit their analyses to regions for which they have full phylogenetic coverage and see if the results still stand?

RESPONSE: We appreciate the reviewer's concerns regarding the phylogenetic resolution in our study. We agree that highly resolved species-level phylogenies are extremely important for questions about trait evolution and estimating diversification rates (e.g., Rabosky, 2015). However, several studies (e.g., Cadotte et al., 2008; Cadotte et al., 2009; Cadotte, 2015; Li et al., 2019; Qian & Jin, 2021) have previously evaluated the resolution of phylogenies and its ability to detect non-

random community patterns. The consensus is that imperfect resolution below the family level has a minimal effect on detection of regional patterns and statistical significance. This is because the assembled communities usually contain distantly related species across different families or genera that traverse deep nodes. Thus, fine-tuning the relative distance among the small values between species in the same genus will have a minor impact on the calculation of MPD which always includes distantly related species. There is no current evidence that more precise estimations of phylogenetic relationships at the genus level can alter community assembly inferences, but there are many analyses showing that results are very robust to changes in distances at the genus level (see Li et al., 2019 and Qian & Jin, 2021 as examples).

In the revised manuscript, we now clarify that the reason why we also used a genus-level phylogeny with polytomies for all species is that some species in our mega tree species were added as polytomies of their genera or family, while other species are fully resolved at the species level (Lines 332-336). This could introduce potential biases when examining latitudinal and climatic gradients in naturalized-alien-to-native phylogenetic distances. We are open to removing this analysis if the reviewer believes it would improve the clarity and conciseness of the paper.

In response to the suggestion of the reviewer, we attempted to perform an analysis using regions with complete molecular phylogeny coverage, but unfortunately we could not find such regions. As we proposed above, using a finer phylogeny might have a minor effect on the calculation of the MPD for a certain region, but for studies across regions, we believe that the overall global patterns we observed would not be significantly altered by improved phylogenetic resolution. However, we agree with the reviewer that constructing better-resolved phylogenies across a broad latitudinal gradient is an important future research direction. We have highlighted this point in the discussion section of the revised manuscript (Lines 252-261).

Line 332: The rationale for limiting a species pool to species with known economic uses is not clear, especially when the focal taxon itself does not have a known use, and needs to be better fleshed out/justified. Why not just limit this pool to species that have been introduced somewhere?

RESPONSE: Ideally, we would indeed use a pool of species that have been introduced to a region (irrespective of whether the species have become naturalized there). As such data unfortunately do not exist, we used different species pools that make different assumptions about which species might have been introduced to a region. For example, the global non-native species pool assumes that all species have been introduced everywhere, whereas the global naturalized species pool assumes that all species that have already become naturalized somewhere have been introduced everywhere. We also recognize the importance of economic plants, which have a higher likelihood of being introduced to other regions for cultivation by humans and often become naturalized. To capture this aspect, we created a species pool that combines all species with known economic uses with all species that have become naturalized somewhere. We believe this is a very important null model, as was also pointed out by Reviewer 1, and we therefore would like to keep it in the manuscript. We have now provided a more detailed explanation of the rationale behind this null model in the manuscript (Lines 388-392).

Line 365: The modeling of species distributions with such coarse data are troubling. 250 km² can comprise extreme amounts of environmental heterogeneity – e.g., coastal invaders can become inland invaders of the desert or alpine invaders. Perhaps some comparisons between these crude models and those fit to actual point occurrence data could be compared for taxa with enough data to support this approach – many (if not most) invasive species have large amounts of occurrence data available. Also, all model parameters should be provided to ensure reproducibility.

RESPONSE: We fully agree that these species distribution models (SDM) used in our study are very crude. While it is true that some widely naturalized (i.e., invasive) species have extensive occurrence records, this is not the case for all naturalized species at a global scale. Limiting the analysis to well-recorded species could introduce biases when studying the latitudinal gradient on a global scale. The purpose of our SDMs is not to precisely predict suitable habitats for naturalized species but to calculate their mean MPD to native species in regions with potentially suitable climates. Therefore, the key question is whether the mean MPD calculated using the coarse data we employed is biased compared to using well-resolved occurrence data.

To address this concern, we made a comparison between mean MPD calculated using our ‘rough’ SDMs and SDMs based on point-occurrence data for a subset of species. We randomly selected 300 naturalized species from the global naturalized alien flora, and then used MaxEnt to predict their global potential distribution based on point occurrence data from the Global Biodiversity Information Facility (GBIF; <https://www.gbif.org/>).

Firstly, we used the R package `rgbif` (Chamberlain & Boettiger, 2017; Chamberlain et al., 2023) to download the point occurrences of these species from GBIF. We kept the occurrences with geographical coordinates. Then, we used the R package `CoordinateCleaner` (Zizka et al., 2019) to clean the geographic coordinates by excluding occurrences that are most likely incorrect. These include coordinates in the ocean (not on islands), zero coordinates, coordinate-country mismatches, coordinates assigned to centroids of countries and provinces, coordinates within cities, outlier coordinates, coordinates assigned to biodiversity institutions, and coordinates of fossil specimens. Then, we used the same six bioclimatic variables as used for the other analyses (including mean diurnal range, temperature seasonality, maximum temperature of warmest month, precipitation seasonality, precipitation of wettest quarter, and precipitation of driest quarter, Lines 416-423) from WorldClim at 2.5 arc-min resolution. We removed occurrences with a coordinate uncertainty greater than 4.5 km (~2.5 arc-min), and removed pseudo-duplicate records, so that there is at most one record per 2.5 arc-min grid cell. Similar to our analysis described in the main text, we first excluded species with fewer than 15 occurrences, and we then ran the MaxEnt model (using the R package `dismo`, Hijmans et al., 2021) separately for each naturalized species. Here, we selected 20% of the data for testing the model, and kept only those models with AUC values greater than 0.7. We then used the maximum training sensitivity plus specificity threshold to test whether a grid cell is suitable for a species. Finally, we obtained the global potential distribution of 272 of the 300 selected naturalized species.

We then compared the mean MPD values based on the SDMs using the coarse distribution data and those using point-occurrence data for the 272 species. The analysis revealed a strong correlation between these MPD values (Pearson's $r = 0.99$, Response Letter Figure 4), indicating that our analyses based on the coarse data, available for all naturalized species, provide a reliable estimation of MPD.

Furthermore, we have included all the necessary information on model parameters in the revised manuscript to ensure reproducibility (Lines 413-441). We appreciate the valuable input of the reviewer and thank them for bringing these points to our attention.

Response Letter Figure 4 | Comparison of phylogenetic distances calculated using two different datasets. For each naturalized species, phylogenetic distances were calculated as the mean value of the mean pairwise phylogenetic distances (MPD) of the non-native species to the native species in the regions with a suitable climate. These results are based on a subset of 272 species for which we found 15 or more point-occurrences in GBIF (out of an initial subset of 300 species). The value along the x-axis is calculated based on the coarse distribution used in our main text, and the value along the y-axis is calculated based on the point-occurrence data from GBIF.

Qian, H., Zhang, Y., Zhang, J. and Wang, X. (2013), Phylogenetic structure of angiosperm tree assemblages. *Global Ecology and Biogeography*, 22: 1183-1191. <https://doi.org/10.1111/geb.12069>

LI, X. H., ZHU, X. X., Niu, Y., & Sun, H. (2014). Phylogenetic clustering and overdispersion for alpine plants along elevational gradient in the Hengduan Mountains Region, southwest China. *Journal of Systematics and Evolution*, 52(3), 280-288.

Escoriza, D., & Ruhí, A. (2014). Macroecological patterns of amphibian assemblages in the Western Palearctic: Implications for conservation. *Biological Conservation*, 176, 252-261.

Park, D. S., Feng, X., Maitner, B. S., Ernst, K. C., & Enquist, B. J. (2020). Darwin's naturalization conundrum can be explained by spatial scale. *Proceedings of the National Academy of Sciences*, 117(20), 10904-10910.

Kusumoto, B., Kubota, Y., Shiono, T., & Villalobos, F. (2021). Biogeographical origin effects on exotic plants colonization in the insular flora of Japan. *Biological Invasions*, 23(9), 2973-2984.

Zu, K., Zhang, C., Chen, F., Zhang, Z., Ahmad, S., & Nabi, G. (2023). Latitudinal gradients of angiosperm plant diversity and phylogenetic structure in China's nature reserves. *Global Ecology and Conservation*, 42, e02403.

Yang, Y., Wei, C., Xiao, L., Zhong, Z., Li, Q., Wang, H., & Wang, W. (2022). Effects of urbanization on woody plant phylogenetic diversity and its associations with landscape features in the high latitude northern hemisphere region, Northeast China. *Science of The Total Environment*, 838, 156192.

Cui, Y. C., Song, K., Guo, X. Y., van Bodegom, P. M., Pan, Y. J., Tian, Z. H., ... & Da, L. J. (2019). Phylogenetic and functional structures of plant communities along a spatiotemporal urbanization gradient: Effects of colonization and extinction. *Journal of Vegetation Science*, 30(2), 341-351.

Dawson, W., Moser, D., Van Kleunen, M., Kreft, H., Pergl, J., Pyšek, P., ... & Essl, F. (2017). Global hotspots and correlates of alien species richness across taxonomic groups. *Nature Ecology & Evolution*, 1(7), 0186.

Van Kleunen, M., Dawson, W., Essl, F., Pergl, J., Winter, M., Weber, E., ... & Pyšek, P. (2015). Global exchange and accumulation of non-native plants. *Nature*, 525(7567), 100-103.

RESPONSE: We thank the reviewer for providing these helpful references.

References used in the responses:

Cadotte, M. W. (2015). Phylogenetic diversity–ecosystem function relationships are insensitive to phylogenetic edge lengths. *Functional Ecology*, 29(5), 718-723. doi:10.1111/1365-2435.12429

Cadotte, M. W., Cardinale, B. J., & Oakley, T. H. (2008). Evolutionary history and the effect of biodiversity on plant productivity. *Proceedings of the National Academy of Sciences*, 105(44), 17012-17017. doi:10.1073/pnas.0805962105

Cadotte, M. W., Hamilton, M. A., & Murray, B. R. (2009). Phylogenetic relatedness and plant invader success across two spatial scales. *Diversity and Distributions*, 15(3), 481-488. doi:10.1111/j.1472-4642.2009.00560.x

Chamberlain, S., Barve, V., Mcglinn, D., Oldoni, D., Desmet, P., Geffert, L., & Rambaut, K. (2023). rgbif: Interface to the Global Biodiversity Information Facility API. Retrieved from <https://CRAN.R-project.org/package=rgbif>

Chamberlain, S., & Boettiger, C. (2017). R Python, and Ruby clients for GBIF species occurrence data. *PeerJ PrePrints*. Retrieved from <https://doi.org/10.7287/peerj.preprints.3304v1>

Hijmans, R. J., Phillips, S., Leathwick, J., & Elith, J. dismo: Species Distribution Modeling. R

package version 1.3-5 (2021). Retrieved from <https://CRAN.R-project.org/package=dismo>

Li, D., Trotta, L., Marx, H. E., Allen, J. M., Sun, M., Soltis, D. E., . . . Baiser, B. (2019). For common community phylogenetic analyses, go ahead and use synthesis phylogenies. *Ecology*, 100(9), e02788. doi:10.1002/ecy.2788

Qian, H. & Jin, Y. (2021). Are phylogenies resolved at the genus level appropriate for studies on phylogenetic structure of species assemblages? *Plant Diversity*, 43, 255-263. doi:10.1016/j.pld.2020.11.005

Rabosky, D. L. (2015). No substitute for real data: a cautionary note on the use of phylogenies from birth–death polytomy resolvers for downstream comparative analyses. *Evolution*, 69, 3207-3216. doi:[10.1111/evo.12817](https://doi.org/10.1111/evo.12817)

Schemske, D. W., Mittelbach, G. G., Cornell, H. V., Sobel, J. M. & Roy, K. (2009). Is there a latitudinal gradient in the importance of biotic interactions? *Annual Review of Ecology Evolution and Systematics*, 40, 245-269. doi:10.1146/annurev.ecolsys.39.110707.173430

Thuiller, W., Gallien, L., Boulangeat, I., De Bello, F., Münkemüller, T., Roquet, C. & Lavergne, S. (2010). Resolving Darwin's naturalization conundrum: a quest for evidence. *Diversity and Distributions*, 16, 461-475. doi:10.1111/j.1472-4642.2010.00645.x

van Kleunen, M., Dawson, W., Essl, F., Pergl, J., Winter, M., Weber, E., . . . Pyšek, P. (2015). Global exchange and accumulation of non-native plants. *Nature*, 525, 100-103. doi:10.1038/nature14910

van Kleunen, M., Pyšek, P., Dawson, W., Essl, F., Kreft, H., Pergl, J., . . . Winter, M. (2019). The Global Naturalized Alien Flora (GloNAF) database. *Ecology*, 100(1), 2. doi:10.1002/ecy.2542

van Kleunen, M., Xu, X. Y., Yang, Q., Maurel, N., Zhang, Z. J., Dawson W, . . . Fristoe, T. S. (2020). Economic use of plants is key to their naturalization success. *Nature Communications*, 11, 3201. doi:10.1038/s41467-020-16982-3

Weigelt, P., König, C., & Kreft, H. (2020). GIFT - A Global Inventory of Floras and Traits for macroecology and biogeography. *Journal of Biogeography*, 47(1), 16-43. doi:10.1111/jbi.13623

Zizka, A., Silvestro, D., Andermann, T., Azevedo, J., Duarte Ritter, C., Edler, D., . . . Antonelli, A. (2019). CoordinateCleaner: Standardized cleaning of occurrence records from biological collection databases. *Methods in Ecology and Evolution*, 10(5), 744-751. doi:10.1111/2041-210X.13152

REVIEWERS' COMMENTS

Reviewer #1 (Remarks to the Author):

With this revision, all my comments have been addressed to my complete satisfaction.

From my perspective it also seems that the comments provided by the other reviewer have been addressed very thoroughly, with new analyses and major changes in the main text. In cases of disagreement, the authors provided much information and well-founded argument why this is the case.

I am looking forward to seeing this manuscript published.

Reviewer #2 (Remarks to the Author):

I appreciate the authors' efforts to address reviewer comments. I believe the manuscript is stronger with the added methodological details and discussion of limitations. I also commend the authors' decision to share most of the data underlying their analyses. It is unfortunate that some of the sources the authors assemble these data from, however, are not fully open. Though I do not agree with all the methodological choices made here others have employed similar approaches at times. The 'regions' that form the unit of analysis (which are not defined, I think – how were these regions delimited?) here vary greatly in their size and environmental heterogeneity. While I understand this is likely because species checklists are usually only available for countries, this is not ideal and may bias the results. I recommend that the authors include analyses, figures, and tables presented in the response document as supplements and refer to them in the main text to facilitate understanding of how and why certain choices were made. It may also be good to acknowledge that results could differ with better data/different methods along with this information (e.g., better phylogenetic coverage, inclusion of more data from species-rich regions in southeast Asia and Africa, more specific range data) – I do not think such acknowledgments detract from the findings of the study.

Responses to the comments

Responses to the Reviewer 1's comments

With this revision, all my comments have been addressed to my complete satisfaction.

From my perspective it also seems that the comments provided by the other reviewer have been addressed very thoroughly, with new analyses and major changes in the main text. In cases of disagreement, the authors provided much information and well-founded argument why this is the case.

I am looking forward to seeing this manuscript published.

RESPONSE: We express our gratitude to the reviewer for this very positive evaluation and valuable feedback throughout the revisions.

Responses to the Reviewer 2's comments

I appreciate the authors' efforts to address reviewer comments. I believe the manuscript is stronger with the added methodological details and discussion of limitations. I also commend the authors' decision to share most of the data underlying their analyses. It is unfortunate that some of the sources the authors assemble these data from, however, are not fully open.

RESPONSE: We are grateful for the reviewer's positive assessment of the manuscript and the recognition of the enhancements made through our revisions. Your valuable feedback has undoubtedly played a pivotal role in enhancing the quality of our work. While a small portion of GIFT database are not publicly available due to data sharing restrictions, we would like to emphasize that all the original data utilized in our study, encompassing the comprehensive lists of 219,520 native and 9,531 naturalized alien plant species spanning 487 regions, are openly accessible via Figshare (<https://doi.org/10.6084/m9.figshare.20055611.v5>).

Though I do not agree with all the methodological choices made here others have employed similar approaches at times. The 'regions' that form the unit of analysis (which are not defined, I think – how were these regions delimited?) here vary greatly in their size and environmental heterogeneity. While I understand this is likely because species checklists are usually only available for countries, this is not ideal and may bias the results.

RESPONSE: In accordance with the reviewer's insightful input, we have made the necessary clarification that the regions examined in our study pertain to administrative units, comprising countries, states, provinces, and islands (Line 288). We agree with the reviewer that the variation in size and environmental diversity among these regions could introduce a potential bias to the results. To address this concern, we conducted supplementary analyses by incorporating region area as a covariate in our statistical model. These

supplementary analyses unequivocally demonstrated that the variation in region size had a modest impact on our results and did not compromise the integrity of our primary findings (Supplementary Fig. 8) (Lines 304-306).

I recommend that the authors include analyses, figures, and tables presented in the response document as supplements and refer to them in the main text to facilitate understanding of how and why certain choices were made. It may also be good to acknowledge that results could differ with better data/different methods along with this information (e.g., better phylogenetic coverage, inclusion of more data from species-rich regions in southeast Asia and Africa, more specific range data) – I do not think such acknowledgments detract from the findings of the study.

RESPONSE: We are glad that the reviewer found these additional analyses helpful. To address this recommendation, we have referenced the completeness analysis in the main text (Lines 514-526) and providing the results in Supplementary Table 4. To ensure the supplementary materials remain focused and easily navigable for readers, we have retained Response Letter Figures 1-4 in the response document. As these figures primarily address specific details discussed with the reviewer and might not be of general significance to all readers, we believe this approach maintains clarity without overwhelming the supplementary materials. These response materials are also fully accessible to the readers, thus ensuring transparency. Furthermore, we have taken your advice and expanded our discussion to acknowledge the potential implications of improved data quality on our findings (Lines 253-255, 266-267).